# The Proteomic Landscape of *CTNNB1* Mutated Low-Grade Early-Stage Endometrial Carcinomas

**DOI:** 10.3390/cells14211676

**Published:** 2025-10-27

**Authors:** Alvaro Lopez-Janeiro, Emilia Brizzi, Ignacio Ruz-Caracuel, Raluca Alexandru, Carlos de Andrea, Alberto Berjón, Laura Yebenes, Marta Mendiola, Victoria Heredia-Soto, Ana Montero-Calle, Rodrigo Barderas, Vivian de los Rios, Andrés Redondo, Alberto Pelaez-Garcia, David Hardisson

**Affiliations:** 1Department of Pathology, Clínica Universidad de Navarra, 31008 Pamplona, Spain; alopezj@unav.es (A.L.-J.); ralexandru@unav.es (R.A.); ceandrea@unav.es (C.d.A.); 2Department of Pathology, Hospital Universitario Puerta del Mar, 11009 Cádiz, Spain; mariae.brizzi.sspa@juntadeandalucia.es; 3Pathology Department, Hospital Universitario Ramón y Cajal & Instituto de Investigación Biomédica Ramón y Cajal (IRYCIS), CIBERONC, 28034 Madrid, Spain; ignacio.ruz@salud.madrid.org; 4Department of Pathology, Hospital Universitario La Paz, 28046 Madrid, Spain; alberto.berjon@salud.madrid.org (A.B.); laura.yebenes@salud.madrid.org (L.Y.); marta.mendiola@salud.madrid.org (M.M.); 5Molecular Pathology and Therapeutic Targets Group, La Paz University Hospital (IdiPAZ), 28046 Madrid, Spain; 6Center for Biomedical Research in the Cancer Network (Centro de Investigación Biomédica en Red de Cáncer, CIBERONC), Instituto de Salud Carlos III, 28029 Madrid, Spain; victoriam.heredia@salud.madrid.org; 7Translational Oncology Research Laboratory, Hospital La Paz Institute for Health Research (IdiPAZ), 28046 Madrid, Spain; andres.redondos@uam.es; 8Chronic Disease Programme (UFIEC), Instituto de Salud Carlos III, Majadahonda, 28220 Madrid, Spain; ana.monteroc@isciii.es (A.M.-C.); r.barderasm@isciii.es (R.B.); 9CIBER Frailty and Healthy Aging, 28029 Madrid, Spain; 10Centro de Investigaciones Biológicas, CSIC, 28040 Madrid, Spain; vrios@cib.csic.es; 11Department of Medical Oncology, Hospital Universitario La Paz, 28046 Madrid, Spain; 12Faculty of Medicine, Universidad Autónoma de Madrid, 28029 Madrid, Spain; 13Proteomics Unit (UCCTs), Instituto de Salud Carlos III, Majadahonda, 28220 Madrid, Spain

**Keywords:** proteomics, endometrial carcinoma, Wnt pathway, immune microenvironment

## Abstract

**Highlights:**

**What are the main findings?**

**What is the implication of the main finding?**

**Abstract:**

Endometrial carcinoma is the most frequent gynecologic malignancy in western countries. In recent years, mutations in *CTNNB1* have been associated with worse prognosis in low-risk carcinomas. However, there is a lack of understanding of the proteomic implications of *CTNNB1* mutations in this type of tumor. In this study, we performed shotgun proteomics using Formalin-Fixed Paraffin-Embedded (FFPE) tissue samples of *CTNNB1* mutated and wild-type low-risk endometrial carcinomas. A publicly available proteomic and transcriptomic database was used to validate results. Differential protein expression and Gene Set Enrichment Analysis revealed dysregulation of pathways associated with cell keratinization, immune response modulation, and intracellular calcium regulation. *CTNNB1* mutated tumors showed immune dysregulation at multiple levels including cytokine secretion, cell adhesion, and lymphocyte activation. These results were supported by tissue multiplex immunofluorescence analysis, demonstrating reduced CD8 tumor-infiltrating lymphocytes and different immune spatial interaction patterns. Intracellular calcium dysfunction was associated with key transcript dysregulation. We found an increased expression of CAMK2A and ROR2, suggesting a potential role for non-canonical Wnt pathway activation in *CTNNB1* mutated tumors.

## 1. Introduction

Endometrial cancer is the leading gynecologic cancer and the fourth leading cancer diagnosis in Europe and North America [1]. Given that endometrial carcinomas frequently present as low-grade tumors diagnosed in early stages, disease-associated mortality remains low. However, a subset of low-grade early-stage tumors can demonstrate aggressive behavior [2]. The introduction of endometrial cancer molecular profiles has significantly improved our understanding of tumor biology and provided a valuable prognostic tool [3,4]. However, most cases belonging to this specific subset of endometrial carcinomas are classified as no specific molecular profile (NSMP) or Mismatch Repair-deficient (dMMR), which are associated with variable disease outcomes. As a result, there are ongoing efforts to further refine the molecular classification of these carcinomas.

In recent years, there has been a surge in molecular biomarkers that can potentially explain the variable tumor behavior of low-grade early-stage endometrial carcinomas [5,6,7,8]. Of these, *CTNNB1* gene mutations have demonstrated to be associated with an adverse prognosis [9,10]. Tumors harboring *CTNNB1* mutations show increased risk of relapses, and distant metastases. *CTNNB1* gene encodes the β-catenin protein, which is involved in canonical Wnt signaling pathway. This pathway is involved in various biological processes including tumor growth, migration, and metastasis [11,12,13]. In addition, this pathway has been associated with other relevant aspects of tumor biology like anti-tumor immune response evasion [14] or epithelial-to-mesenchymal transition (EMT) [15]. Indeed, Wnt pathway dysregulation is associated with several malignancies, including colorectal carcinoma, head and neck tumors, and infrequent pediatric neoplasms [16,17,18]. However, the implications of these mutations in low-grade early-stage endometrial tumors remain incompletely understood.

Proteomics has long been considered a powerful tool to interrogate the molecular landscape of malignancies [19]. Recently, researchers have used proteomics to characterize endometrial carcinomas [20,21,22]. The use of this comprehensive technology has facilitated the discovery of relevant protein expression markers associated with low-grade endometrial cancer biology [21]. Recently, there have been efforts to optimize the performance of this type of “omic” technologies using Formalin-Fixed Paraffin-Embedded (FFPE) Tissue samples [23]. Therefore, combining shotgun proteomics with FFPE samples, which are readily available in the clinical setting, is a promising approach to improve biomarker discovery and our understanding of tumor biology. In the present work, we aim to harness the power of shotgun proteomics to further explore the biological impact of *CTNNB1* mutations using FFPE samples from low-grade early-stage endometrial carcinomas.

## 2. Materials and Methods

### 2.1. Patient Selection and CTNNB1 Mutation Determination

For proteomic analysis tumor samples from low-grade (FIGO Grade G1 or G2) early-stage (FIGO stage 2018 I or II) endometrioid endometrial carcinomas (EEC) were selected from the archives of Hospital Universitario La Paz. To be eligible, patients must have been treated with hysterectomy between 2003 and 2015. Formalin-Fixed Paraffin-Embedded (FFPE) tissue blocks obtained from hysterectomy specimens with at least 50% viable tumor cell enrichment were used for proteomic analysis. To interrogate *CTNNB1* mutation status, we performed *CTNNB1* exon 3 sequencing as previously published [10]. We selected FFPE tissue samples containing at least 50% of viable tumor cells. DNA from these samples was obtained by QIAamp FFPE tissue kit (Qiagen, Hilden, Germany) and used for PCR and Sanger sequencing. CTNNB1 exon 3, encompassing the region of GSK-3β phosphorylation site, was amplified with these specific primers (5′-3′): GATTT-GATGGAGTTGGACATGG and TGTTCTTGAGTGGAAGGACTGAG. Only pathogenic (COSMIC scores > 0.7) variants were considered.

### 2.2. Protein Extraction and Processing

Tissue samples were sectioned using a microtome (10 µm thick), transferred into 1.5 mL tubes, and deparaffinized by incubation with xylol for 5 min at 56 °C, followed by sequential ethanol washes (100%, 95%, 80%, and 50%). The samples were boiled by heating for 2 h at 95 °C (500 rpm). Tissue samples were air-dried, followed by resuspension in EasyPep Lysis Buffer. Protein concentration of the samples was measured using Qubit Protein Assay. Sample preparation for reduction, alkylation, trypsin digestion, and cleanup were performed according to the EasyPep protocol.

### 2.3. TMT Labeling and LC-MS

TMT labeling was performed as previously described [24]. In brief, TMT reagents (0.8 mg) were dissolved in aceto-nitrile (40 μL), of which 20 μL was added to the peptides (50 μg). Peptide quantification was performed by Thermo Scientific™ Pierce™ Quantative Colorimetric Peptide assay to determine peptide concentration before LC-MS loading and TMT experiments. Normalized digested samples were labeled by incubation at room temperature for 1 h (500 RMP) with TMT10plex reagents. The reaction was quenched with hydroxylamine to a final concentration of 0.3% (*v*/*v*). TMT-labeled samples were pooled at a 1:1 ratio across all 10 samples. For each experiment, the pooled sample was vacuum-centrifuged to near dryness. Pierce High pH Reversed-Phase Fractionation Kits were used to fractionate TMT-labeled digest samples into eight fractions by an increasing aceto-nitrile step-gradient elution. Fractions were dried in a vacuum centrifuge and resuspended in 0.1% formic acid prior to LC-MS analysis.

Afterwards, the fractions were mixed into six fractions, dried under vacuum, then resuspended in 12 µL and quantified by fluorimetry (QuBit) as previously described [25]. One μg of each sample was subjected to nano-Liquid Chromatography coupled to Electrospray Ionization Tandem Mass Spectrometry using a nano Easy 100 (Thermo Fisher Scientific, Waltham, MA, USA) coupled online to an Q Exactive mass spectrometer (Thermo Fisher Scientific) [23]. The mass spectrometry proteomics data have been deposited in the ProteomeXchange Consortium via the PRIDE [26] partner repository with the dataset identifier PXD068004.

### 2.4. Validation Cohort

Proteomic, transcriptomic, CIBERSORT cell deconvolution scores and whole exome sequencing data from the CPTAC-endometrial carcinoma consortium was accessed from the LinkedOmics repository [27]. Patients diagnosed with low-grade early-stage endometrioid endometrial carcinomas were selected. Samples without tumor purity information were also excluded from the analysis. *CTNNB1* mutation status was obtained from the whole exome sequencing curated data, including all mutation types of *CTNNB1*. Normalized proteomic data and transcriptomic data as well as CIBERSORT data were used in further analyses.

### 2.5. Statistical Analysis

To identify differentially expressed proteins, we first analyzed differential expressions in the discovery cohort. We calculated the Log_2_FC, as well as *t*-test, followed by q value correction using the q value Bioconductor R package (version 2.36.0). Proteins with a q value below 0.1 were subjected to further screening using the validation cohort. Further analyses were conducted on proteins that exhibited consistent differential expression profiles in both the discovery and validation cohorts, and those were among the top 10% of proteins that were either upregulated or downregulated in both groups.

Differentially expressed proteins were further analyzed using Gene Set Enrichment Analysis (GSEA). We obtained pathway gene sets using the msigdbr R package (version 7.5.1). First, we identified pathways that contained at least one protein of interest from the Gene Ontology and Kegg pathway collections. We then performed GSEA using the fgsea Bioconductor R package (version 1.30.0). Proteomic data from both the discovery and validation cohorts, along with transcriptomic data from the validation cohort were used to calculate the Normalized Enrichment Score for all the selected pathways. Pathway Enrichment Scores were subsequently screened by filtering out pathways that showed inconsistent up- or downregulation profiles across the three datasets. Finally, pathways that ranked among the top 15% upregulated or downregulated in all three datasets were considered for further validation studies.

### 2.6. Quantitative PCR Analysis

RNA samples were isolated from representative FFPE tissue blocks from low-grade, early-stage primary EEC with and without *CTNNB1* mutation using QIAGEN RNeasy Mini kit, following manufacturer instructions. RNA was quantified by spectrophotometry. Quantitative real-time polymerase chain reaction was performed in an Mx3005p (Agilent, Santa Clara, CA, USA) using the SYBR^®^ Green Quantitative RT-qPCR Kit (Merck, Darmstadt, Germany). Expression of target RNAs was normalized using ACTB, GADPH, and PPIA genes as internal controls [28]. A list of primers for VDAC1, VDAC2, Wnt5A, ROR2, and CAMK2A can be found in Appendix A. Results were analyzed using the CT method [29].

### 2.7. Tissue-Based Analyses

To perform quantitative immunofluorescence and immunohistochemical analysis, we constructed Tissue Micro arrays (TMAs) as previously reported [30]. Each TMAs contained two 1.2 mm cores per cancer sample obtained from hysterectomy specimen FFPE tissue blocks. Prior to analysis, TMA cores showing significant artifacts (tissue detachment) or an absence of tumor tissue were excluded.

Multiplex immunofluorescence was performed as previously published by our group [30]. In brief, each TMA section was subjected to iterative rounds of antibody staining against CK (1:150, AE1/AE3; Novus Biologicals, Littleton, CO, USA), CD8 (1:150, 4B11; Bio-Rad, Hercules, CA, USA), FOXP3 (1:50, 236A/E7; Abcam, Cambridge, UK) and CD68 (1:75, PG-M1; Dako-Agilent, Santa Clara, CA, USA). Antibody binding was followed by tyramide signal amplification (TSA) visualization with fluorophores Opal-690, Opal-540, Opal-570, and Opal-620. Nuclei were counter-stained with DAPI (Akoya Biosciences, Marlborough, MA, USA). Immunofluorescence imaging and spectral unmixing were performed using the PhenoImager platform (Akoya Biosciences). Cell segmentation was performed using the SimpleSeg Bioconductor R package (version 1.4.1). Afterwards marker expression (quantile 97.5) was calculated for each cell and image channel, and cell expression was normalized using the COMBAT algorithm implemented in the mxnorm R package (version 1.0.3). Using protein expression data, cell phenotyping was conducted. To this end, we first identified positive/negative thresholds for each marker and phenotyping decision trees were designed to identify tumor cells, CD8 T cells, macrophages, and FOXP3 Treg cells. Cells not fulfilling criteria for any of the above-mentioned cell types were classified as “other”. Total tissue area was approximated using cell coordinate information. To identify tumor and stromal compartments, tumor masks were constructed based on the spatial location of cells classified as tumor cells. To this end, we first filtered out spatially isolated tumor cells using a DBSCAN algorithm implemented in the dbscan R package (Version 1.1.12). Clustered cells were then used to approximate a concave hull and build tumor masks using the sf R package (Version 1.0.18).

After cell-density quantification, we conducted spatial interaction analyses using SpatialExperiment (version 1.12.0) and imcRtools (version 1.8.0) R packages. We analyzed the spatial interaction/repulsion pattern of CD8, macrophage, Tregs, and tumor cells. To this end, we first constructed a spatial neighborhood graph, using Delaunay triangulation for every cell–cell potential interaction. Homotypic and heterotypic cell interactions were quantified by dividing graph edges between graph nodes. Afterwards, images were classified based on their cell–cell interaction patterns using a k-means algorithm with k = 4. Spatial interaction clusters were compared across *CTNNB1* mutated and wild-type (WT) samples.

To analyze ROR2 protein expression, TMA sections were stained with a monoclonal anti-human ROR2 antibody (1:100, ROR2 2535-2835, QED bioscience, San Diego, CA, USA). Sections were counter-stained using hematoxylin (Leica Biosystems, Nussloch, Germany). An expert pathologist who was blind to *CTNNB1* mutation status evaluated the samples. The average percentage of stained cells was calculated for every tissue core. Both nuclear and cytoplasmic staining were considered irrespective of staining intensity.

To review the presence of squamous differentiation, a pathologist who was blind to *CTNNB1* mutation status was requested to review the hematoxilyn and eosin-stained tissue slides. First, whole tissue slides samples belonging to the discovery cohort used in the proteomic analysis were evaluated. Next, to evaluate the presence of squamous differentiation in the samples from the validation cohort, digitalized HE slides were downloaded from the Cancer Imaging Archive [31]. Images were annotated using very large image viewer (VLIV) [32] by the same pathologist.

## 3. Results

### 3.1. Patient Characteristics of Discovery and Validation Cohorts

The clinicopathological and molecular characteristics of the discovery and validation cohorts have been recently published [10,21,30], and summarized in Table 1. A total of 18 tumors were analyzed in the discovery cohort (12 WT, 6 harboring *CTNNB1* mutations), whereas 60 tumors were included in the validation cohort (37 WT, 23 *CTNNB1* mutated). Both *CTNNB1* WT and *CTNNB1* mutated tumors showed similar clinical and pathological characteristics. However, we observed that *CTNNB1* mutated tumors appeared in slightly older patients in the discovery cohort, with the opposite observance for the validation cohort. No dMMR tumor subtypes were included in the discovery cohort. However, immunohistochemical findings categorized six patients (10% of the validation cohort) as dMMR. Notably, no *POLE* mutations were found in the discovery or validation cohort.

### 3.2. Protein Quantification and Gene Set Enrichment Analysis

FFPE tissue samples from the 18 discovery cohort patients were subjected to shotgun proteomic analysis using a Q Exactive mass spectrometer and TMT10plex. A total of 1618 proteins were identified and quantified. As an initial screening strategy, we performed differential protein expression analysis between WT and *CTNNB1* mutated tumors, resulting in 1032 proteins with a q value below 0.1. The expression pattern of these proteins was further screened in the validation samples. Among these proteins, 934 were identified and quantified in the validation cohort, 474 showed the same up-/downregulation profile, and 30 were among the top 10% of up- or downregulated proteins in both the discovery and the validation cohorts (Figure 1A). The validated proteins displayed diverse biological functions. First, we identified β-catenin (encoded by *CTNNB1* gene) protein to be upregulated in *CTNNB1* mutated samples compared to WT samples. Second, we found several ion channels and amino acid transporters to be dysregulated in *CTNNB1* mutated samples, including VDAC1, VDAC3, and SLC1A5. We also identified several proteases and lysosome-associated proteins to be differentially expressed, like LAMP2, SCARB2, MMP7, GGH, and DPEP1. Additionally, other proteins associated with cell cytoskeletal system and cell adhesion were also dysregulated, including KRT18 and ALCAM. Finally, we also identified several proteins associated with protein translation and endoplasmic reticulum function, likeTMED2, TMED9, TMED10, RPL32, and RPS16.

Given the heterogeneous biological backgrounds of the differentially expressed proteins, we conducted a Gene Set Enrichment Analysis (GSEA) to further investigate the impact of *CTNNB1* mutations on the tumor proteome. To this end, we selected 1444 pathways from the GO and Kegg pathway repositories, which included at least one of our 30 proteins of interest. The Net Enrichment (NE) Score of these pathways was compared across the proteomic data of the discovery and validation sample sets, as well as the transcriptomic data of the validation cohort. We selected pathways that showed an NE score within the highest or lowest 15% across the three datasets. After filtering, 24 pathways were found to be consistently dysregulated (Figure 1B). These could be grossly classified into pathways associated with keratinization or squamous cell differentiation, immune microenvironment shaping, and cellular calcium homeostasis regulation.

### 3.3. CTNNB1 Mutated Tumors Show Increased Squamous Differentiation Capabilities

Several of the differentially activated pathways in *CTNNB1* mutated tumors were related to keratinization and epidermal growth. This finding was consistently observed across proteomic and transcriptomic databases. To further validate this observation, we reviewed the hematoxylin and eosin slides of the validation cohort and annotated the presence of squamous differentiation. We found that squamous differentiation was more frequent in samples harboring *CTNNB1* mutation compared to WT controls (Figure 2). Among the six *CTNNB1* mutated tumors, five (83%) showed squamous differentiation, in contrast to the 58% of WT samples (7 out of 12) (Fisher exact test = 0.6). Similarly, in the validation cohort, among the 36 evaluable CTNNNB1 WT cases, 20 (56%) showed squamous differentiation. Conversely, 17 out of the 23 cases exhibiting *CTNNB1* mutations (74%) showed this morphologic characteristic (Fisher exact test = 0.179).

### 3.4. CTNNB1 Mutated Tumors Show Differences in the Tumor Immune Microenvironment

Most of the differentially activated pathways were associated with immune response regulation. Differentially downregulated pathways were associated with a variety of immune regulation processes.

In this sense, we identified that *CTNNB1* mutated samples were associated with downregulation of IL-1 production pathways. Therefore, to further explore this finding, we analyzed the mRNA expression level of IL1 pathway-related interleukins and receptors using the validation RNAseq dataset. We found that compared to *CTNNB1* WT carcinomas, samples harboring *CTNNB1* mutations showed significantly decreased levels of IL1A (*p*-value = 0.045), IL1B (*p*-value = 0.12), and caspase 1 (*p*-value = 0.037) (Appendix A), while demonstrating similar IL1R1 and IL1R2 expression levels.

Additionally, we observed downregulation of leukocyte transendothelial migration and leukocyte cell–cell adhesion pathways. Consistent with this result, major cell adhesion molecules were found to be downregulated at the transcriptomic level. Specifically, we found that *CTNNB1* mutated carcinomas showed significantly downregulated levels of ITGAL, ITGAM, and ITGB2, which are involved in leukocyte transendothelial migration. Finally, several pathways associated with lymphocyte functions, including T cell receptor (TCR) signaling regulation and lymphocyte activation, were dysregulated. Accordingly, key members of the TCR signaling pathway, like CD3E and NFKB1, were significantly downregulated in *CTNNB1* mutated tumors. Moreover, INF-gamma mRNA expression level also showed a non-significant downregulated profile in *CTNNB1* mutated samples (Appendix A).

To further investigate the differences in the immune profile between *CTNNB1* mutated and WT samples, we assessed different immune populations in the validation cohort deconvoluting the transcriptomic data using the CIBERSORT algorithm. We identified an net enrichment of the myeloid monocyte population in *CTNNB1* mutated samples (average CIBERSORT score 0.02 vs. 0.009, two-sided *t*-test *p*-value = 0.016). On the other hand, CD8 T cells were less abundant in *CTNNB1* mutated tumors compared to WT samples, although this difference did not reach statistical significance (average CIBERSORT score 0.061 vs. 0.09, two-sided *t*-test = 0.16) (Figure 3).

Additionally, we analyzed a large cohort of low-grade early-stage endometrial carcinomas. A total of 303 tissue cores (31 cores from *CTNNB1* mutated and 272 cores from WT tumors) from 176 patients were evaluated using quantitative immunofluorescence. On average, *CTNNB1* mutated tumors exhibited lower CD8 cell density (452 vs. 280 CD8 cells per mm^2^, two-sided *t*-test *p*-value < 0.01). This lower CD8 infiltration in *CTNNB1* mutated samples was consistent across both intra-tumoral and stromal compartments (two-sided *t*-test *p*-value < 0.001 for intra-tumor CD8 TILs and 0.013 for stromal TILs). In addition, we found a slight but non-significant increase in CD68-positive macrophages in the tumoral compartment with no differences in the stromal compartment (Figure 3).

To further understand immune microenvironment differences between *CTNNB1* mutated and wild-type tumors, we analyzed spatial interaction patterns. We observed different patterns of homotypic and heterotypic cell–cell interaction between samples. After clustering, we identified four groups of samples according to cell–cell interaction patterns. The most common pattern of cell–cell interaction was characterized by weak spatial interactions (low spatial clustering overall) and was therefore termed weak interaction patterns. It was the most frequently found pattern in both *CTNNB1* mutated and wild-type samples. However, *CTNNB1* samples were enriched in these spatial interaction patterns, with more than 50% of samples being classified as weakly interactive. The second spatial interaction pattern was enriched in immune cells spatially interacting with tumor cells and tumor cells showing homotypic interactions. Therefore, we labeled these samples as tumor–immune interactions. These samples were equally found in both types of samples. Finally, we identified two additional clusters enriched in homotypic and heterotypic interactions between immune cells. These clusters represented more than 25% of *CTNNB1* WT tumors but were only rarely seen in their mutated counterparts (13%) (Fisher’s exact test *p*-value = 0.46) (Figure 3).

### 3.5. CTNNB1 Mutated Tumors Show Dysregulation of Intracellular Calcium and Calcium-Dependent Wnt Pathway Signaling

Six of the differentially activated pathways were associated with cytosolic calcium homeostasis. We identified anion channels and transporter-related pathways as consistently dysregulated in *CTNNB1* mutated tumors. Given that our differential protein expression analysis identified mitochondrial anion channels VDAC1 and VDAC3 as two of the most upregulated proteins in *CTNNB1* mutated samples, we first validated this upregulation using qPCR analysis. We analyzed discovery cohort samples and observed an increase in VDAC1 mRNA levels in *CTNNB1* mutated samples and a moderate increase in VDAC2 mRNA levels (Figure 4).

We hypothesized that the dysregulation of cytosolic calcium regulation could be involved in Wnt signaling through non-canonical pathways, including Wnt/Ca^2+^. To evaluate this hypothesis, we analyzed the mRNA levels of Wnt/Ca^2+^ related mediators by qPCR on FFPE tissue samples. The primary ligand of Wnt/Ca^2+^ receptor, Wnt5a, was upregulated in *CTNNB1* mutated tumors (Figure 4). The transmembrane signaling transductor ROR2 also showed increased levels in samples with β-catenin mutations (Figure 4). Finally, the Calcium Calmodulin Kinase II, which plays a key role in signal transduction, was also upregulated in *CTNNB1* mutated tumors. To further investigate these findings, we analyzed the ROR2 expression in a large cohort of low-grade early-stage endometrial carcinomas (150 samples from *CTNNB1* wild-type tumors and 15 from mutated counterparts). ROR2 immunohistochemistry revealed increased higher protein expression levels in *CTNNB1* mutated samples. On average, *CTNNB1* mutated tumors showed 33.3% of ROR2 positive cells, while only 13.3% of WT tumor cells showed ROR2 positivity.

## 4. Discussion

β-catenin mutations have been observed in a variety of malignancies. The activation of the Wnt pathway in malignant tumors has been demonstrated to be a critical factor in the promotion of tumor growth, migration, and metastasis [11,13]. Although the prognostic impact of *CTNNB1* mutations in low-grade early-stage endometrial carcinoma has already been described [9,10], the biological mechanisms underlying the aggressive behavior of these tumors remain incompletely understood. In this study, we investigate the proteome of low-grade early-stage endometrial carcinomas to explore the biologic impact of *CTNNB1* mutations.

We observed here a marked dysregulation of cell differentiation processes in *CTNNB1* mutated endometrial tumors. Specifically, tumor harboring mutations in β-catenin gene showed increased activity of squamous differentiation pathways. The formation of immature squamous morules has been previously described in endometrial carcinomas [33,34]. Squamous differentiation is a characteristic observed in various non-squamous epithelial neoplasms, including urothelial and breast carcinomas, and is usually associated with poor prognosis or advanced tumor stages [35,36]. Although there are some conflicting results, the significance of squamous differentiation in endometrial carcinomas has also been proposed as an adverse prognostic factor [37,38]. Our findings support the idea that Wnt pathway activation may contribute to a more aggressive phenotype in low-grade early-stage carcinomas by promoting squamous differentiation.

Additionally, we found that *CTNNB1* mutated tumors are associated with singular immune microenvironment features. Tumor-specific immune response is supposed to influence the biological behavior of carcinomas [39], and previous studies have highlighted the prognostic relevance of immune infiltration profiles in low-grade early-stage carcinomas [30]. Wnt pathway activation has been previously linked with an immunosuppressive microenvironment [14,40]. Tumors showing alterations in Wnt pathway are, on average, less infiltrated by lymphocytes. Furthermore, previous reports have indicated that tumor-infiltrating lymphocytes (TILs) are insuffificiently primed by antigen-presenting cells in *CTNNB1* mutated animal tumor models [40]. Our findings are in line with previously published papers. We have found that the immune microenvironment of low-grade early-stage endometrial carcinomas is characterized by dysregulation of pathways associated with cytokine secretion, leukocyte migration, and lymphocyte activation. Furthermore, these tumors showed an overall decrease in the number of CD8+ lymphocytes and increased myeloid cells. Furthermore, our spatial interaction analyses have revealed that *CTNNB1* mutated tumors exhibit not only a decrease in CD8+-infiltrating lymphocytes but also distinct spatial interaction patterns compared to their WT counterparts. We have seen that *CTNNB1* mutated tumors show lower CD8 homotypic interactions, as well as heterotypic interactions with other immune cells. This diminished spatial interaction pattern could be associated with the lower T cell activation that has been described in these tumors. Overall, these findings support the notion that *CTNNB1* mutated endometrial carcinomas adopt a more aggressive phenotype due to enhanced immune evasion mechanisms. Our results suggest that immunotherapy approaches aimed at low-grade early-stage carcinomas should consider *CTNNB1* mutation status when developing treatment strategies.

Furthermore, we have observed an association between *CTNNB1* mutations and elevated calcium ion transport, and Wnt/Ca^2+^ pathway activity. Canonical Wnt/β-catenin pathway activation is mediated by interaction of Wnt ligands with Frizzled receptors, leading to the translocation of β-catenin into the nucleus. On the other hand, Wnt/Ca^2+^ pathway activation is dependent on Wnt5a signaling, ROR1, and ROR2 activation, and subsequent increase in cytosolic calcium levels [41,42,43]. However, the interdependence between these two pathways remains an area of active research. In the present work, we identified a potential connection between Wnt/β-catenin and Wnt/Ca^2+^ signaling pathways. Our findings are supported by previously published works [44] that have described calcium as a key mediator for β-catenin nuclear entry. These interaction favors the signaling capacity of *CTNNB1* and increases the activity of the Wnt pathway. Furthermore, other authors have found that tumors with *CTNNB1* mutation that show nuclear protein translocation have higher mRNA expression levels of Trop2, a protein known to regulate intracellular calcium levels [45]. As the inhibition of Wnt pathway signaling is a promising therapeutic approach [46], our findings suggest that inhibition of Wnt/Ca^2+^ pathway could improve the therapeutic potential of Wnt signaling inhibition. In addition, recent studies suggest a connection between intracellular calcium regulating protein expression and tumor-immune microenvironment through INF-gamma signaling regulation [47]. This suggests a potential connection between *CTNNB1* mutation, and the immune microenvironment and calcium dysregulation found in the present study. Further studies are needed to validate the biological interplay between these two pathways.

However, there are several limitations that are worth mentioning. The proteome coverage in mass spectrometry experiments performed on FFPE tissue samples is smaller than that obtained using fresh frozen tissue. Moreover, the FFPE samples used in the present work had been stored for several years. Although there is evidence that protein loss in archived FFPE samples is minimal [48,49], given the impact on protein retrieval caused by formalin fixation [50], it is possible that relevant dysregulated proteins may not have been identified in our study. In addition, although we have identified dysregulated pathways through bulk proteomics and transcriptomics, the spatial expression pattern of some of these relevant proteins has not been studied. It is possible that non-tumor cells or even extracellular matrix components partially explain the dysregulation of some of the relevant proteins (like lysosome-associated proteins) found in our study. Further spatial proteomic studies may be required to address this question. Finally, our descriptive proteomic analysis could greatly benefit from further functional experiments. In vivo or in vitro studies could further refine the question of whether protein dysregulation is associated with reduced protein production or protein secretion, and how these dysregulation patterns influence tumor cell behavior and interaction with other cell types.

## 5. Conclusions

The tumor proteome of *CTNNB1* mutated low-grade early-stage carcinomas differs from their WT counterparts. Proteome dysregulation is associated with several key biological pathways including squamous differentiation, immune evasion, and cell calcium homeostasis regulation. Our results support the idea that *CTNNB1* mutations have an impact on the biological behavior of these tumors.

## Figures and Tables

**Figure 1 cells-14-01676-f001:**
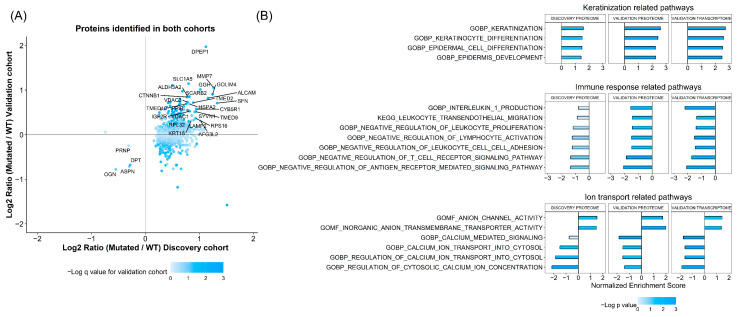
(**A**) Log_2_ Fold change (*CTNNB1* mutated vs. WT) according to discovery and validation cohorts. (**B**) GSEA results across discovery and validation cohorts (including proteome and transcriptome).

**Figure 2 cells-14-01676-f002:**
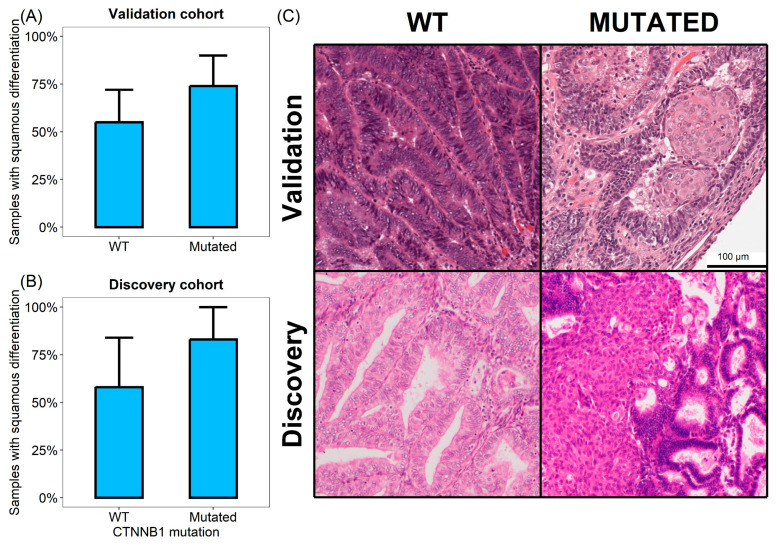
Samples with squamous differentiation in discovery cohort (**A**) and validation cohort (**B**). (**C**) Representative microphotographs demonstrating squamous differentiation in mutated cases but not in WT samples.

**Figure 3 cells-14-01676-f003:**
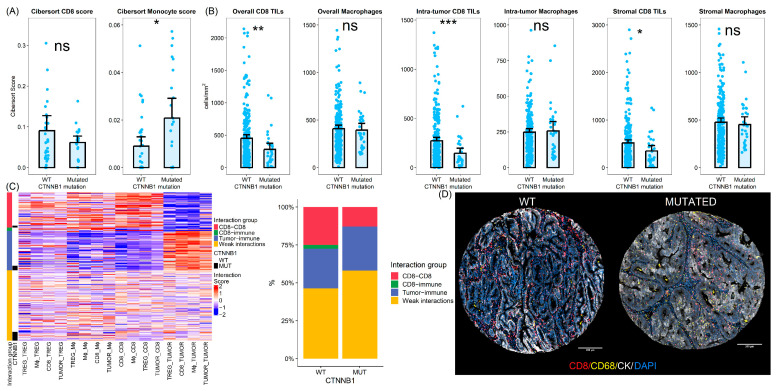
(**A**) Results from the CIBERSORT scores in the validation cohort. (**B**) Multiplex Immunofluorescence analysis in a cohort of low-grade early-stage carcinomas. (**C**) Heatmap and bar plot representing spatial interaction analysis results according to MIF results. (**D**) Representative photographs of *CTNNNB1* wild-type and mutated tumors. ns: not significant. *: *p* < 0.05, **: *p* < 0.01, ***: *p* < 0.001.

**Figure 4 cells-14-01676-f004:**
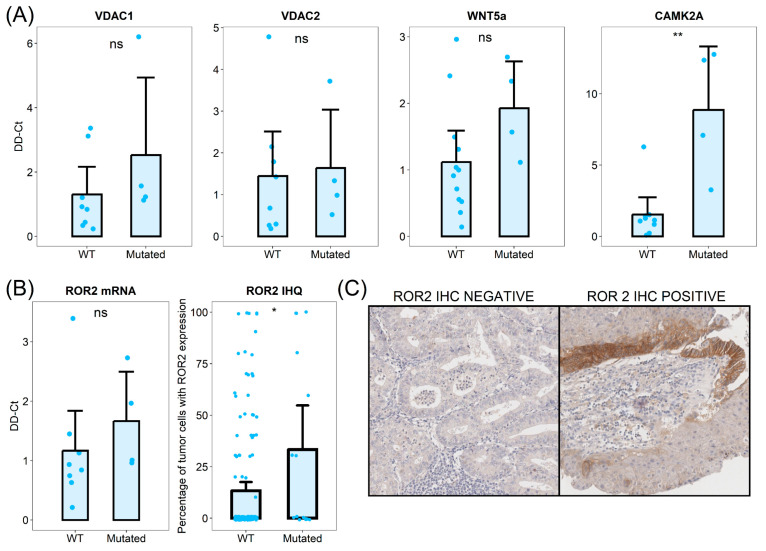
(**A**) Transcript expression analysis between WT and mutated tumors. (**B**) ROR2 mRNA and immunohistochemical analysis between WT and mutated tumors. (**C**) Representative microphotographs representing ROR2 negative and positive tumors. ns: not significant. *: *p* < 0.05, **: *p* < 0.01.

**Table 1 cells-14-01676-t001:** Clinicopathological and molecular characteristics of the discovery and validation cohorts.

		Discovery Cohort		Validation Cohort	
		*CTNNB1* WT	*CTNNB1* MUT	*CTNNB1* WT	*CTNNB1* MUT
**n**		12	6	37 (62%)	23 (38%)
Age (y, mean, p25–p75)		63 (58–67)	73 (69–78)	65 (61–69)	59 (55–65)
FIGO 2018 (n, %)	FIGO IA	5 (42%)	2 (33%)	25 (68%)	19 (83%)
	FIGO IB	7 (58%)	3 (50%)	8 (22%)	3 (13%)
	FIGO II	0	1 (17%)	3 (8%)	1 (4%)
	FIGO I NOS	0	0	1 (3%)	0
Grade (n, %)	Grade 1	9 (75%)	4 (67%)	15 (41%)	16 (70%)
	Grade 2	3 (25%)	2 (33%)	22 (59%)	7 (30%)
LVI (n, %)	Absent	8 (67%)	3 (50%)	31 (84%)	20 (87%)
	Present	4 (33%)	3 (50%)	6 (16%)	3 (13%)
MMR (IHC status)	Proficient	12 (100%)	6 (100%)	4 (11%)	13 (55%)
	Deficient	0	0	6 (16%)	0
	NA	0	0	27 (73%)	10 (45%)
POLE	WT	9 (75%)	3 (50%)	23 (100%)	37 (100%)
	Mutated	0	0	0	0
	NA	3 (25%)	3 (50%)	0	0
RELAPSE (n, %)	Local	2 (17%)	3 (50%)		
	Distant, nodal	4 (33%)	3 (50%)		
	NA	0	0	23 (100%)	37 (100%)

Abbreviations: FIGO: International Federation of Gynecology and Obstetrics, IHC: immunohistochemistry, LVI: lymphovascular invasion, MMR: mismatch repair, MUT: mutated, NA: not available, NOS: not otherwise specified, WT: wild-type.

## Data Availability

The mass spectrometry proteomics data have been deposited in the ProteomeXchange Consortium via the PRIDE partner repository with the dataset identifier PXD068004.

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
