# Peer review of "The Proteomic Landscape of CTNNB1 Mutated Low-Grade Early-Stage Endometrial Carcinomas"

_cells, 2025, doi:10.3390/cells14211676_

Round 1
Reviewer 1 Report
Comments and Suggestions for Authors
Major Comments:
- The manuscript provides in depth analysis of a layered study that starts with a modest profiling experiment and progresses through established validation stages. The first validation stage is the CPTAC consortium. The depth of coverage for the FFPE tissue at 1,618 proteins is very modest for a TMT-labeled and Alkaline Reversed Phase fractionated tissue using a QExactive system. Since this is the data set that forms the foundation of the study the coverage is disappointing.
- This is a well designed study that follows a well established sequence of discovery proteomics to generate new data, validation & comparison to database recoverable data sets and finally large scale tissue array experimentation using targeted imaging.
- This is a well conducted study that utilizes informatic resources including software and data repository very effectively. The authors make useful connections to immune cell type distributions that are pertinent to the mechanistic understanding of endometrial cancer progression.
Minor Comments:
- Fig 2. There is no scale marker in the figure panel 2C. It appears that the scale changes between the WT and the tumor images. They should have a scale marker and that scale should be the same for all images.
- Pp 2, ln 75. “REcently” appears to be a typo.
Reviewer 2 Report
Comments and Suggestions for Authors
This study examines the proteomic landscape of CTNNB1-mutated low-grade early-stage endometrial carcinomas using shotgun proteomics on FFPE tissue samples. The authors analyzed 18 discovery cohort samples (6 CTNNB1-mutated vs 12 wild-type) and validated findings using a public database of 60 samples. They identified 30 differentially expressed proteins and conducted pathway enrichment analysis, revealing dysregulation in keratinization, immune response, and calcium homeostasis pathways. The study suggests that CTNNB1 mutations are associated with increased squamous differentiation, altered tumor immune microenvironment with reduced CD8+ T cell infiltration, and potential activation of non-canonical Wnt/Ca2+ signaling. While these findings provide interesting preliminary observations, several fundamental methodological and interpretational issues warrant careful consideration.
Major Comments
1. A critical limitation that remains unaddressed is the inability to determine cellular origin of the identified proteins. The bulk tissue TMT proteomics approach cannot distinguish whether protein changes originate from tumor cells, immune cells, or stromal components.
2. The use of FFPE samples for TMT proteomics introduces substantial technical limitations that significantly impact data quality and interpretation. Formalin fixation causes extensive protein crosslinking, blocks 30-50% of lysine residues essential for TMT labeling, and reduces protein extraction efficiency to merely 20-40% of fresh tissue yields. These chemical modifications create selective protein loss, with membrane and hydrophobic proteins being particularly underrepresented.
3. The 12-year span of sample collection (2003-2015) introduces time-dependent degradation of proteins that could account for observed differences. The manuscript would greatly benefit from acknowledging these FFPE-specific limitations and discussing how they might create artifactual protein expression differences unrelated to CTNNB1 mutation status.
4. The protein quantification strategy using total protein normalization (50 μg per sample) fails to account for tissue composition differences between samples. Changes in extracellular matrix content, fibrosis, or cellular density could artificially skew the apparent protein expression levels. For example, increased ECM deposition in CTNNB1-mutated tumors would dilute intracellular proteins, making them appear downregulated even if cellular expression remains unchanged. A discussion of how tissue architecture variations might confound the quantitative results would strengthen the interpretation of the data.
5. The validation strategy relies heavily on public database comparison rather than experimental verification of the key proteins. Among the 30 differentially expressed proteins, only ROR2 was validated by immunohistochemistry. Critical findings like VDAC1/3 upregulation and IL-1 pathway downregulation lack protein-level validation with spatial resolution. The authors could enhance their conclusions by acknowledging these validation gaps and proposing specific follow-up experiments needed to confirm cellular sources and functional relevance of these protein changes.
6. The interpretation of secreted proteins presents a conceptual challenge that deserves attention. The study cannot distinguish between intracellular stores and secreted forms of proteins like IL-1 and MMP7. Decreased tissue levels of IL-1 could indicate either reduced production (supporting the immune suppression hypothesis) or increased secretion (suggesting inflammation). This ambiguity fundamentally affects the biological interpretation and potential therapeutic implications. Discussing how secretion dynamics and protein clearance rates influence the observed protein levels would provide a more understanding of the findings.
7. While the authors state that proteomic data has been deposited to ProteomeXchange with identifier PXD068004, this dataset is currently not accessible for review. Data availability is crucial for transparency and reproducibility in proteomics research. The authors should ensure that reviewers have access to the raw data during the review process
Minor Comments
1. Line 76: "Formalyn" should be corrected to "Formalin"
2. Lines 69-70: "the biological **implications** of these mutations... remains incompletely understood" should read "implications remain"
3. Line 75: "REcently" contains an erroneous capital E and should be "Recently"
4. Throughout manuscript: Inconsistent use of "p value" and "p-value" - should be standardized
5. Line 318: "two-sided t-test = 0.016" should read "two-sided t-test p = 0.016"
6. Table 1 footnote: The asterisk explanation format is unclear and should be reorganized for clarity
Round 2
Reviewer 2 Report
Comments and Suggestions for Authors
The authors have adequately addressed the major concerns raised in the previous review. The revised manuscript now appropriately discusses the limitations associated with FFPE sample preparation, tissue composition, and the bulk proteomic approach. The inclusion of clarifying statements in the discussion section, together with the provision of access to the raw proteomic dataset (PXD068004), adds substantial clarity and credibility to the work. The overall manuscript quality has been significantly improved and is acceptable in its current form.